# TCR-EML: Explainable Model Layers for TCR-pMHC Prediction

[1]**Jiarui Li**, [1]**Zixiang Yin**, [1]**Zhengming Ding**, [2]**Samuel J. Landry**, [1][*]**Ramgopal R. Mettu**

[1] Department of Computer Science, Tulane University
[2] Department of Biochemistry and Molecular Biology, Tulane University School of Medicine
`{jli78, zyin, hsmith19, zding1, landry, rmettu}@tulane.edu`
[*]Corresponding author.
**https://tcreml.jiarui.li/**

## ABSTRACT

T cell receptor (TCR) recognition of peptide-MHC (pMHC) complexes is a central component of adaptive immunity, with implications for vaccine design, cancer immunotherapy, and autoimmune disease. While recent advances in machine learning have improved prediction of TCR-pMHC binding, the most effective approaches are black-box transformer models that cannot provide a rationale for predictions. Post-hoc explanation methods can provide insight with respect to the input but do not explicitly model biochemical mechanisms (e.g. known binding regions), as in TCR-pMHC binding. "Explain-by-design" models (i.e., with architectural components that can be examined directly after training) have been explored in other domains, but have not been used for TCR-pMHC binding. We propose explainable model layers (TCR-EML) that can be incorporated into protein-language model backbones for TCR-pMHC modeling. Our approach uses prototype layers for amino acid residue contacts drawn from known TCR-pMHC binding mechanisms, enabling high-quality explanations for predicted TCR-pMHC binding. Experiments of our proposed method on large-scale datasets demonstrate competitive predictive accuracy and generalization, and evaluation on the TCR-XAI benchmark demonstrates improved explainability compared with existing approaches.

## 1 INTRODUCTION

For the adaptive immune system, T cells are essential for detecting and responding to antigens from pathogens such as viruses, bacteria, and cancer cells (Joglekar & Li, 2021), as well as in autoimmune contexts. The final step of T cell activation involves binding between a peptide presented by the Major Histocompatibility Complex (pMHC) and the T cell receptor (TCR). The specificity of this interaction is the foundation of T cell-mediated immunity and is a major focus of research in both therapeutic development and the study of immune mechanisms. A detailed understanding of T cell response is critical for designing vaccines that provide durable immunity and for developing effective personalized cancer treatments (Rojas et al., 2023; Poorebrahim et al., 2021).

Prediction of T cell receptor and peptide-major histocompatibility complex (TCR-pMHC) binding, a key step of T cell response, remains a central challenge in quantitative immunology and adaptive immunity (Hudson et al., 2023). Both unsupervised and supervised learning algorithm approaches have been explored (Hudson et al., 2023; 2024). Unsupervised methods cluster TCR sequences using similarity metrics such as TCRdist3 (Mayer-Blackwell et al., 2021) applied to complementarity-determining regions (CDRs), without requiring binding labels or epitope information (e.g., GLIPH2 (Huang et al., 2020)). The resulting clusters are then used to support experimental analysis (Hudson et al., 2024). Supervised methods utilize large TCR-pMHC datasets (Hudson et al., 2023) from sources including VDJdb (Bagaev et al., 2020), McPAS-TCR (Tickotsky et al., 2017), and IEDB (Vita et al., 2019), and employ deep learning models such as MixTCRpred (Croce et al., 2024), NetTCR2.2 (Jensen & Nielsen, 2023), TULIP (Meynard-Piganeau et al., 2024), and EGM (Li et al., 2025a).

These models, however, operate as black boxes, and their lack of explainability hinders biological insight into T cell recognition. To address this, post-hoc explanation methods (Kenny et al., 2021) such as QCAI (Li et al., 2025b) has been proposed. These methods demonstrate that deep models can capture mechanisms of TCR-pMHC binding and generate rational predictions (Li et al., 2025b;a).

However, post-hoc explanations are not always faithful and have limitations when applied to black-box models (Rudin, 2019).

To address these challenges we have developed an explain-by-design prediction head for TCR-pMHC modeling that can be used with PLM backbones (e.g., ProteinBERT (Brandes et al., 2022), ESM-1b Rives et al. (2021), and ESM-2 (Lin et al., 2023)). Our approach makes these widely used models interpretable, without retraining the entire architecture. Our design for TCR-pMHC binding incorporates contact prototypes that can be interrogated after training to reveal mechanistic insights. We evaluate our approach on large-scale TCR-pMHC binding

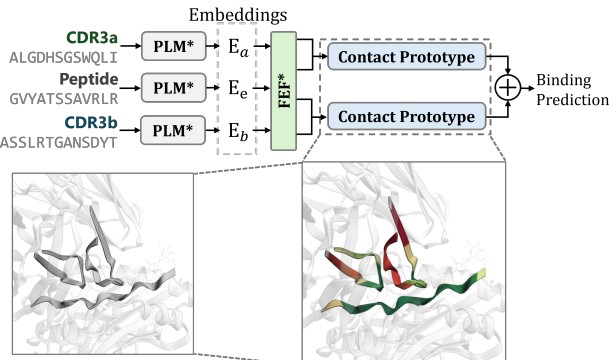

Figure 1: The explainable model layers include a Feature Enhancement and Fusion (FEF) block and contact prototype layers, which not only predict TCR-pMHC binding but also provide contact scores corresponding to contact distances. In the absence of experimental TCR-pMHC structures, the contact prototype illuminates TCR-pMHC binding patterns.

datasets for accuracy and generalization, and on the TCR-XAI benchmark (Li et al., 2025b) for explainability, where it achieves superior performance to existing models.

## 2 OUR APPROACH

Our design consists of two components: (1) feature enhancement and fusion, and (2) contact prototype layers. These components can be directly attached to PLM backbones, which provide embeddings for CDR3a, CDR3b, and peptide sequences, denoted as $E_a \in \mathbb{R}^{N \times d}$, $E_b \in \mathbb{R}^{N \times d}$, and $E_e \in \mathbb{R}^{N \times d}$, where $N$ is the maximum sequence length and $d$ is the embedding dimension.

### 2.1 FEATURE ENHANCEMENT AND FUSION

Since different pre-trained PLMs adopt diverse architectures and are developed for general-purpose protein modeling, there is no guarantee that the embeddings of CDR3a, CDR3b, and peptide are effectively fused. To address this, we introduce a feature enhancement and fusion (FEF) module that integrates multiple cross-attention layers, motivated by the design principles of Explanation-Guided Model (EGM) (Li et al., 2025a), a method using post-hoc analyses to guide transformer model design. Formally, we denote cross-attention from $a$ to $b$ as $\mathcal{A}(Q = a, K, V = b)$, where $a$ serves as the query and $b$ as the key and value. Guided by EGM, we first derive cross-fused representations of CDR3s using $E_{a \to b} = \mathcal{A}(Q = E_a, K, V = E_b)$ and $E_{b \to a} = \mathcal{A}(Q = E_b, K, V = E_a)$. Subsequently, the peptide embeddings are fused with $E_{a \to b}$ and $E_{b \to a}$ to obtain enriched features for TCR-pMHC modeling: $E_{e \to a \to b} = \mathcal{A}(Q = E_e, K, V = E_{a \to b}), E_{a \to b \to e} = \mathcal{A}(Q = E_{a \to b}, K, V = E_e), E_{e \to b \to a} = \mathcal{A}(Q = E_e, K, V = E_{b \to a}), E_{b \to a \to e} = \mathcal{A}(Q = E_{b \to a}, K, V = E_e)$.

### 2.2 CONTACT PROTOTYPE LAYERS

We design prototype-based layers to explicitly model contacts between TCR and pMHC using the fused features from FEF. These layers estimate residue contacts between CDR3a and peptide, and between CDR3b and peptide, respectively. For each pair, the fused embeddings are denoted as $E_1 \in \mathbb{R}^{N \times d}$ and $E_2 \in \mathbb{R}^{N \times d}$. The contact prototype layers take them as inputs and calculate the contact area between these chains. Inspired by the cross-attention mechanism, we model contact distance through similarity $S = \left( E_1 \cdot E_2^\top / \|E_1\| \cdot \|E_2\| \right) \cdot \tau \in [0, 1]^{N \times N}$, where $\tau \in \mathbb{R}^+$ is a trainable temperature parameter. Higher similarity corresponds to shorter contact distance.

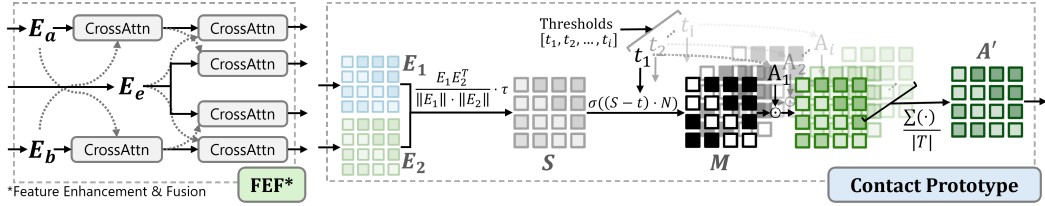

Figure 2: Overview of the our explainable model layers for TCR-pMHC binding prediction. The Feature Enhancement and Fusion (FEF) block integrates information between TCR chains and TCR-peptide pairs. Contact prototype layers model residue-level contact areas and distances between CDR3 regions and the peptide.

We introduce a set of thresholds $T = [t_0, t_1, ..., t_{|T|}]$ to filter potential contacts, where $|T|$ is the number of thresholds and each $t_i \in [0, 1]$. For threshold $t_i$, residues with similarity greater than $t_i$ are considered to be in contact. To ensure differentiability, we approximate this contact filter as: $M_i = \sigma((S - t_i) \cdot N) \in [0, 1]^{N \times N}$, where $\sigma$ is the sigmoid function. Following the principle that shorter distances generally imply larger contact areas, we define contact areas under each threshold $t_i$ using $A = \text{softmax}(T) \in [0, 1]^{|T|}$. The contact area is then computed as $A' = \left( \sum_{i=1}^{|T|} \sqrt{M_i \odot A_i} \right) / |T| \in [0, 1]^{N \times N}$, where $A'_{k,j}$ denotes the contact area between residues $k$ and $j$, and $\odot$ is element-wise product. The overall contact area between the embeddings $E_1$ and $E_2$ is defined as $w_{1,2} = \frac{1}{N^2} \sum_{k=1}^{N} \sum_{j=1}^{N} A'_{k,j} \in [0, 1]$, where $N^2$ is the maximum possible contact area. For notation clarity, we define a contact prototype function $f : \mathbb{R}^{N \times d} \times \mathbb{R}^{N \times d} \to [0, 1]$ with $f(E_1, E_2) = w_{1,2}^c$. Finally, the contact areas between CDR3a and peptide, and CDR3b and peptide, are given by $w_{a,e} = f(E_{e \to b \to a}, E_{b \to a \to e})$ and $w_{b,e} = f(E_{e \to a \to b}, E_{a \to b \to e})$. The final contact score for a TCR-pMHC pair is summarized as $\hat{y} = (w_{a,e} + w_{b,e})/2$. Since $w$ serves as a direct indicator of TCR-pMHC binding, it is optimized using a class-weighted cross-entropy loss that accounts for the positive-to-negative ratio in the training data: $\mathcal{L} = \mathcal{H}_{\text{CE}}(\hat{y}, y)$, where $\mathcal{H}_{\text{CE}}$ denotes the cross-entropy loss and $y \in \{0, 1\}$ is the ground-truth binding label.

## 3 RESULTS AND DISCUSSION

In this section, We present and discuss the results of evaluation using standard metrics (i.e., ROC-AUC, accuracy) as well binding region hit rate (BRHR) designed to assess explainability.

### 3.1 ROC-AUC ANALYSIS

| ROC-AUC | ProteinBERT | | ESM-1b | | ESM2-150M | | MixTCRpred | TULIP |
|---|---|---|---|---|---|---|---|---|
| Top-$k$ | Linear | Ours | Linear | Ours | Linear | Ours | | |
| 100 | 0.772 | **0.999** | 0.900 | **0.982** | 0.713 | **0.985** | 0.906 | 0.821 |
| 150 | 0.675 | **0.895** | 0.795 | **0.854** | 0.633 | **0.860** | 0.773 | 0.706 |
| 200 | 0.625 | **0.792** | 0.716 | **0.759** | 0.593 | **0.765** | 0.698 | 0.648 |

Table 1: ROC-AUC comparison at Top-100, Top-150, and Top-200 peptides. Columns report results for selected PLM backbones (ProteinBERT (Brandes et al., 2022), ESM-1b (Rives et al., 2021), and ESM-2 (Lin et al., 2023)) with either a linear classifier or our method, with MixTCRpred (Croce et al., 2024) and TULIP (Meynard-Piganeau et al., 2024) included as reference baselines.

To assess the performance and generalization ability of our method, we report the ROC-AUC with the maximum false positive rate restricted to 0.1, which is a standard way for TCR-pMHC binding prediction evaluation (Nielsen et al., 2024)[1]. The evaluation is conducted on the compiled test set with only peptides not observed during training. For each backbone model, we summarize the top-$k$ peptides with the highest ROC-AUC in Table 1. As shown in Table 1, all PLM backbones combined with our method outperform TULIP and MixTCRpred. In particular, ProteinBERT with our method

---

[1]Please check Appendix A.2 for train and test dataset information.

achieves an ROC-AUC of 99.9% on the Top-100 epitopes, representing improvements of approximately 9% and 17% over MixTCRpred and TULIP, respectively. Compared to linear classifiers, our method improves performance by about 24% on average with ProteinBERT and ESM-2 backbone. With ESM-1b, our method performs 4-8% higher ROC-AUC than linear classifiers.

## 3.2 EVALUATION OF THE TCR-XAI BENCHMARK

| $a \rightarrow b$ | ProteinBERT | ESM-1b | ESM2-150M | MixTCRpred | TULIP |
|---|---|---|---|---|---|
| Peptide→CDR3a | 0.839 | 0.842 | 0.897 | 0.718 | 0.702 |
| Peptide→CDR3b | 0.842 | 0.877 | 0.850 | 0.723 | 0.634 |
| CDR3a→Peptide | 0.736 | 0.812 | 0.773 | 0.795 | 0.798 |
| CDR3b→Peptide | 0.790 | 0.813 | 0.792 | 0.675 | 0.646 |

Table 2: Binding Region Hit Rate ($t$=0.25) across different PLM backbones for Peptide→CDR3a, Peptide→CDR3b, CDR3a→Peptide, and CDR3b→Peptide, where $a \rightarrow b$ denotes $a$ contacts to $b$.

We evaluate the explanation quality of the contact prototypes using the TCR-XAI benchmark (Li et al., 2025b). The Binding Region Hit Rate (BRHR) (Li et al., 2025b) quantifies the proportion of true binding residues, defined by structural proximity, that are correctly identified by an explanation method. [2] Table 2 presents the BRHR results across PLM backbones for Peptide→CDR3a, Peptide→CDR3b, CDR3a→Peptide, and CDR3b→Peptide interactions, where $a \rightarrow b$ denotes contacts from $a$ to $b$. For peptide to CDR3 interactions, all backbones with contact prototype layers achieve BRHR values above 0.73. For peptide to CDR3 interactions, BRHR exceeds 0.83 across all PLMs with contact prototypes. These results indicate that our contact prototype layers provide reliable residue-level explanations of CDR3-peptide interactions in TCR-pMHC binding prediction.

## 3.3 CASE STUDY

To illustrate the practical use of our model, we present a case study on a self-antigen associated with rheumatoid arthritis. Specifically, we analyze the HLA-DR4-bound citrullinated peptide vimentin-64cit59-71 (PDB: 8TRR) (Loh et al., 2024). Using ProteinBERT in combination with our TCR-EML, which demonstrates superior performance and generalization, we visualized contact distance weights between the peptide and TCR. To summarize peptide interactions, we computed an integrated contact scores by averaging the contact scores of CDR3s.

As shown in Figure 3, the contact distances predicted by our method closely match the experi-

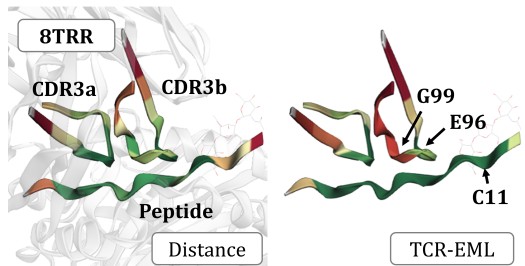

Figure 3: Predicted versus experimental peptide-CDR3 contact distances for HLA-DR4-bound vimentin-64cit59-71 (PDB: 8TRR) (Loh et al., 2024). TCR-EML predictions closely match experimental contacts, highlighting model explainability.

mentally determined distances, with only minor deviations. The BRHR ($t = 0.25$) achieves 1.0 for peptide and CDR3a, and 0.67 for CDR3b. Notably, in CDR3b, TCR-EML correctly identifies the E96 contact region but misses G99, which leads to the lower BRHR for this region. On the peptide side, our method assigns a high weight to C11, which is not a true contact site. These minor discrepancies notwithstanding, the results indicate that TCR-EML provides high-quality, interpretable explanations that faithfully capture biologically relevant contact regions.

## 4 CONCLUSION

In summary, we introduce TCR-EML, explainable model layers for TCR-pMHC binding prediction that enhance both interpretability and generalization to unseen epitopes. TCR-EML can be integrated with pre-trained protein language models, converting them into explainable predictors. Across experiments, TCR-EML augmented PLMs outperform linear classifiers and state-of-the-art methods. Case studies and evaluations on the TCR-XAI benchmark demonstrate that TCR-EML

---

[2]Please check Appendix A.3 and A.4 for TCR-XAI benchmark and BRHR introduction.

yields accurate, biologically meaningful explanations consistent with experimental structures. Because of the contact prototype design of TCR-EML, future work can utilize experimental structures to regularize prototype learning, guiding the model toward more accurate TCR-pMHC modeling and more robust contact pattern explanations.

### ACKNOWLEDGMENTS

The authors acknowledge support from the Harold L. and Heather E. Jurist Center of Excellence for Artificial Intelligence at Tulane University. This work was also supported by the National Institutes of Health (U54-CA260581) through the Tulane University COVID Antibody and Immunity Network (TUCAIN); Tulane SOM Pilot Funding for "MHCII Pathway Processing of SARS-CoV-2 Spike"; and the Lavin-Bernick Faculty Grant Proposal Research and Scholarly Activities Support for "Research Trainee Support for Modeling Antigen Processing and HLA Immunopeptidomics".

## MEANINGFULNESS STATEMENT

Although machine learning models achieve high accuracy in TCR-pMHC and other biological predictions, their black-box nature limits reliability and biological insight. Our approach learns biologically meaningful representations of TCR-pMHC binding and bases predictions on them, improving interpretability and revealing contact patterns even when experimental TCR-pMHC structures are scarce.

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

## A  APPENDIX

### A.1  INTRODUCTION TO T CELL ACTIVATION

CD8+ T cells are activated through the MHCI pathway, whereas CD4+ T cells are activated through the MHCII pathway. Epitope prediction for CD8+ T cells has achieved notable success, while mechanisms of CD4+ T cell response remain less well understood. The CD4+ T cell response can be viewed as a two-stage recognition process. In the first stage, antigens are processed by antigen-presenting cells (APCs) and loaded onto MHCII molecules, which are subsequently presented on the APC surface (Davis & Bjorkman, 1988; Neefjes et al., 2011). In the second stage, TCRs recognize these pMHC complexes, initiating the T cell response. TCR recognition is mediated by the $\alpha$ and $\beta$ domains, each composed of variable (V), joining (J), and constant (C) regions, with the $\beta$ chain also containing a diversity (D) region (Bosselut, 2019). Accurate prediction of T cell responses therefore requires modeling both antigen processing and TCR-pMHC binding (Peters et al., 2020; Nielsen et al., 2020).

Early computational work in epitope prediction emphasized peptide-MHCII binding using allele-specific machine learning methods (Nielsen et al., 2020), with tools including NetMHCpan (Hoof et al., 2009; Nielsen et al., 2007) and NetMHCcons (Karosiene et al., 2012). More recently, antigen processing has been modeled with the Antigen Processing Likelihood (APL) algorithm (Mettu et al., 2016; Bhattacharya et al., 2023; Li et al., 2024a;b; Charles et al., 2022), which accounts for structural features that determine which peptides are presented by MHCII molecules.

## A.2 TRAINING DATASET AND TEST DATASET WITH UNSEEN EPITOPES

To train and evaluate our model layers, we constructed a TCR-pMHC dataset comprising 349,716 paired samples of TCR alpha and beta chains and 2,316 unique peptides, which spans *Homo sapiens* and *Mus musculus*. Among the samples, 95.7% correspond to MHC-I and 4.3% to MHC-II. It was compiled from VDJdb (Bagaev et al., 2020), TCR-McPAS (Tickotsky et al., 2017), IEDB (Vita et al., 2019), TBAdb Zhang et al. (2020), and 10x Genomics (10x Genomics, 2022). For all sources, we retained only samples that provided CDR3a, CDR3b, and peptide sequences. Any non-standard characters, irregular notations, or missing residues were discarded in the amino acid sequences. Negative samples were generated by directly sampling negative pairs for the 10x Genomics dataset, and for other datasets by randomly shuffling TCR and pMHC pairs. For each epitope, negative samples were generated at a ratio of 4:1 relative to positive samples. Finally, the dataset was split into training and test sets using a 95:5 ratio. The test set contains 15,503 samples spanning 288 epitopes that do not appear in the training data. To construct the evaluation set, we computed the Levenshtein distance between each pair of peptides. We then sampled a comparable number of peptides whose minimal pairwise distance exceeded different thresholds from 1 to 9, ensuring that all selected epitopes were excluded from and distinct from those in the training dataset.

## A.3 TCR-XAI BENCHMARK

Li et al. (2025b) introduced a benchmark to quantitatively assess explanation quality using residue-level contacts between TCR and pMHC, derived from 274 structural samples. For each sample, residue-level distances were computed in two ways: (1) from each CDR residue to the nearest atom in the peptide, and (2) from each peptide residue to the nearest atom in any CDR region. Smaller distances indicate stronger interactions and are treated as ground-truth for evaluating explanation methods.

## A.4 BINDING REGION HIT RATE

The Binding Region Hit Rate (BRHR) (Li et al., 2025b) quantifies the proportion of true binding residues, defined by structural proximity, that are correctly identified by an explanation method. More concretely, for a chosen percentile threshold $t$, we compare the top $t$ fraction of residues ranked by contact scores and the top $t$ fraction of residues ranked by pairwise distance (between peptide and chosen TCR chain). A residue is counted as a *hit* if is in both sets (i.e., it is considered important for prediction and has close structural proximity). For our experiments, we use $t = 0.25$ because it is the most restrict threshold but ensuring each case has at least one contact residue. The hit rate is computed per sequence type for each positive sample, and the final BRHR is reported as the average over the TCR-XAI benchmark.

