# OpenReview forum: "TCR-EML: Explainable Model Layers for TCR-pMHC Prediction"
_ICLR.cc/2026/Workshop/LMRL — ICLR 2026 Workshop LMRL Poster_

### Official Review · Reviewer_3kJg · 2026-02-22
**TCR-EML: Promising Explainable Architecture Requires Implementation/Design Details**

**Rating:** 5
**Confidence:** 3

**Review:**

Summary
The authors propose TCR-EML, a prediction head designed to integrate with pre-trained Protein Language Models (pLMs) for T Cell Receptor and peptide-MHC (TCR-pMHC) binding prediction. The architecture enhances and fuses pLM embeddings for CDR3$\alpha$, CDR3$\beta$, and peptide sequences via cross-attention modules (Feature Enhancement and Fusion block) and maps gated learned similarities to structural contact areas (Contact Prototype Layers). The authors term this an "explain-by-design" approach due to its explicit biological intention. To evaluate effectiveness, the authors report compelling TCR-pMHC binding performance in terms of partial ROC-AUC (max FPR 0.1) on top-k peptides compared to linear probing and existing baselines (MixTCRpred and TULIP). To validate explainability, the method achieves higher Binding Region Hit Rates (BRHR) than baselines across different pLM backbones.

Strengths
- Relevance to Workshop Themes: The "explain-by-design" approach aligns well with the LMRL workshop's goal of learning meaningful representations of life. The results suggest that designing prediction heads with strong connections to biological intent (e.g., contact maps) yields better performance than generic approaches. This encourages the community to consider how to obtain meaningful, interpretable representations in computational biology.
- Modularity and Simplicity: The FEF and contact prototype modules are conceptually intuitive and appear easy to integrate into different pLM backbones without retraining the entire model. This generality lowers the barrier for adoption.
- Explicit Interpretability: Unlike post-hoc methods, the model provides direct contact scores grounded in biochemical intuition, which are faithful by construction.

Weaknesses and Questions
1. Metric Reporting and Performance Claims:
The paper reports ROC-AUC with the maximum false positive rate restricted to 0.1 (partial AUC). However, Table 1 only reports results for the Top-100, Top-150, and Top-200 peptides. This represents less than 10% of the peptide space and may inflate perceived performance.
- Suggestion: Please report the mean ROC-AUC or PR-AUC across all peptides to provide a more holistic view of performance.
- Concern: The partial AUC of 0.999 (ProteinBERT Top-100) is exceptionally high. Appendix A.2 states negative samples were generated by randomly shuffling TCR-pMHC pairs before the train-test split. Please clarify if this shuffling strategy risks creating trivially easy negatives or data leakage (e.g., if similar negatives appear in both sets).

2. Architecture Comparisons and Ablations:
While the method is compared to linear probing and two baselines (MixTCRpred, TULIP), there is no ablation study isolating the contributions of the Feature Enhancement and Fusion (FEF) block versus the Contact Prototype layers. It would be helpful to see how simpler universal function approximators (e.g., MLP, concatenation + self-attention) perform as a head. Additionally, an ablation study removing either the FEF or Contact layers would clarify which component drives the performance gains versus the explainability benefits. (Note: Given the Tiny Paper track, this is a minor request but would strengthen the claims signficantly).

3. Implementation and Training Details:
Critical implementation details are missing, which hinders reproducibility. For example,
were the pLM backbones frozen or fine-tuned? What were the optimization settings (learning rate, batch size, optimizer, training iterations, etc)? What hardware was used and what was the training time overhead compared to a linear head? Importantly, are the gating thresholds T learnable parameters or predetermined hyperparameters?

4. Case Study Analysis:
In the case study (Section 3.3, PDB: 8TRR), the authors candidly note that the model misses the G99 contact region in CDR3β and falsely assigns high weight to C11 on the peptide. While honest, the manuscript lacks a biochemical hypothesis for these failures. Was this due to sequence scarcity in the training data, structural anomalies, or limitations in the contact prototype formulation? A brief discussion would add depth. Also, is this case study from test or training data? It would be helpful to show more concrete examples especially for the ones with low BRHR from the current approach but high from existing baselines.

5. Clarification on Explainability Claims:
Without clear explanations, the introduction critiques that post-hoc explanations have limitations but does not deeply analyze why cross-attention is inherently more useful here. In fact, pos-hoc analyses (e.g. attention map, attribution analyses) would help clarify what cross-attention learns. It would be helpful to briefly elaborate on why the specific cross-attention design in FEF is necessary compared to standard self-attention mechanisms in the context of TCR-pMHC interaction.

6. Typos and Textual Errors:
Section 3.2: The text states: "For peptide to CDR3 interactions, all backbones with contact prototype layers achieve BRHR values above 0.73. For peptide to CDR3 interactions, BRHR exceeds 0.83..." The second instance should likely read "For CDR3 to Peptide interactions" to match the rows in Table 2. There are also several minor typos throughout (e.g., "languange", "piptide", "achtecture") that should be proofread.

---

### Official Review · Reviewer_xRhY · 2026-02-24
**Well-motivated and technically sound Explain-by-Design approach for TCR-pMHC binding prediction**

**Rating:** 7
**Confidence:** 4

**Review:**

## Summary Of The Paper

This paper introduces TCR-EML, a modular explain-by-design prediction head for T cell receptor (TCR) and peptide-MHC (pMHC) binding prediction. The core contribution is a two-component architecture: (1) a Feature Enhancement and Fusion (FEF) module based on cascaded cross-attention layers that enriches representations of CDR3a, CDR3b, and peptide embeddings from any pre-trained protein language model (PLM); and (2) contact prototype layers that explicitly model residue-level contacts between CDR3 regions and the peptide.

The contact prototype layers estimate binding via a normalised cosine similarity matrix:

    S = ( E_1 * E_2^T / (||E_1|| * ||E_2||) ) * tau  in [0,1]^{N x N}

where tau is a trainable temperature parameter. Soft contact masks are computed as:

    M_i = sigmoid( (S - t_i) * N )

and aggregated into contact areas A' using a softmax-weighted sum over thresholds T = [t_0, ..., t_{|T|}]. The final binding score y_hat = (w_{a,e} + w_{b,e}) / 2 is the average contact area between CDR3a-peptide and CDR3b-peptide pairs, and is optimised directly with class-weighted cross-entropy loss.

The method is evaluated on a large dataset of 349,716 TCR-pMHC pairs across 2,316 peptides, with a held-out test set of unseen epitopes. Results demonstrate that TCR-EML augmented PLMs outperform linear classifiers and state-of-the-art methods (MixTCRpred, TULIP) in ROC-AUC, and achieve superior Binding Region Hit Rate (BRHR) on the TCR-XAI benchmark for explainability.

---

## Strengths And Weaknesses

### Strengths

**(1) Genuinely novel application of explain-by-design principles to a critical immunology problem.**
While prototype-based interpretable models have been explored in computer vision and NLP, their application to TCR-pMHC binding is new and timely. The framing is well-grounded in Rudin (2019)'s argument against post-hoc explanations for high-stakes decisions, in general making a principled and appropriate motivation for the architectural choice.

**(2) Biologically meaningful design.**
The contact prototype formulation is not merely a technical add-on; it is directly grounded in the known biochemistry of TCR-pMHC recognition. The decision to model CDR3a-peptide and CDR3b-peptide contacts separately, and to aggregate them symmetrically, reflects the biological structure of the interaction. This is a genuine strength over generic attention-based explanations.

**(3) Modular and backbone-agnostic design.**
The ability to attach TCR-EML to any PLM backbone (ProteinBERT, ESM-1b, ESM-2) without retraining the entire architecture is a significant practical advantage. The consistent improvements across all three backbones (Table 1) suggest the method is robust and not overfitted to a particular PLM's representation space.

**(4) Strong quantitative results.**
ProteinBERT + TCR-EML achieves ROC-AUC of 0.999 on Top-100 epitopes, representing approximately 9% and 17% improvements over MixTCRpred and TULIP respectively. The improvements are consistent across Top-100, Top-150, and Top-200 splits, and the test set design is rigorous and well-described.

**(5) Superior explainability performance.**
The BRHR results in Table 2 consistently exceed both MixTCRpred and TULIP across all four interaction directions (Peptide->CDR3a, Peptide->CDR3b, CDR3a->Peptide, CDR3b->Peptide) and all three PLM backbones. The case study on vimentin-64cit59-71 (PDB: 8TRR) provides a compelling qualitative validation that the model's contact predictions align with experimentally determined structures.


### Weaknesses

**(1) The contact similarity formulation conflates similarity with physical distance in a non-trivial way.**
The core design choice (using cosine similarity in embedding space as a proxy for physical residue contact distance) is not explicitly justified. High cosine similarity between PLM embeddings of two residues does not necessarily correspond to close spatial proximity in a protein complex; it may instead reflect biochemical or evolutionary similarity. The authors assert "higher similarity corresponds to shorter contact distance" but provide no ablation or analysis to validate this correspondence. A discussion or empirical validation of this assumption would substantially strengthen the technical foundation of the work.

**(2) Only a single case study for qualitative validation.**
Section 3.3 presents one case study on a rheumatoid arthritis-associated self-antigen (PDB: 8TRR). While informative, a single case study is insufficient to draw broad conclusions about the model's biological interpretability. Additional case studies (ideally spanning different MHC alleles, peptide lengths, and disease contexts) or a more systematic structural validation across the TCR-XAI benchmark would strengthen the explainability claims.

**(3) BRHR threshold choice (t=0.25) lacks sufficient justification.**
The authors note in Appendix A.4 that t=0.25 is chosen because it is "the most restrictive threshold but ensuring each case has at least one contact residue." This is a practical justification but not a principled one, since results at other thresholds are not reported, making it unclear whether the performance advantage holds across threshold values or is specific to t=0.25. Reporting BRHR across a range of thresholds (e.g., t = 0.1, 0.25, 0.5) would provide a more complete picture of explanation quality.

**(4) The FEF module's contribution is not isolated.**
The paper evaluates TCR-EML (FEF + contact prototypes) against linear classifiers and existing baselines, but does not ablate the FEF module independently. It is therefore unclear whether the performance gains stem primarily from the contact prototype layers (the novel contribution) or from the FEF cross-attention module (which is adapted from EGM, Li et al. 2025a). An ablation comparing: (i) PLM + linear head, (ii) PLM + FEF + linear head, and (iii) PLM + FEF + contact prototypes, would cleanly isolate the contribution of each component.

**(5) Class imbalance handling is underspecified.**
The training data uses a 4:1 negative-to-positive ratio, and the loss uses class-weighted cross-entropy. However, the specific class weights used are not reported, nor is it discussed how sensitive the method is to this choice. Given that negative sample generation via random shuffling is known to introduce biases in TCR-pMHC datasets, this deserves more explicit treatment.

**(6) Limited comparison to the most recent state of the art.**
The paper compares primarily against MixTCRpred and TULIP. EGM (Li et al., 2025a), a closely related work by the same research group, is cited in the methodology but not included as a comparison baseline in Table 1. Including EGM as a baseline would clarify the marginal contribution of the explain-by-design approach over the post-hoc explanation approach on which part of the architecture is based.

## Questions

1) Could the authors provide empirical evidence or analysis supporting the claim that cosine similarity in PLM embedding space correlates with physical residue contact distance in TCR-pMHC complexes? For instance, a correlation analysis between the learned similarity scores S and experimentally determined distances from the TCR-XAI structural data would be informative.

2) Could the authors report an ablation study separating the contribution of the FEF module from the contact prototype layers? Specifically: how does PLM + FEF + linear head compare to PLM + FEF + contact prototypes in both ROC-AUC and BRHR?

3) Could the authors report BRHR at additional threshold values (e.g., t = 0.1, 0.5) to demonstrate that the explainability advantage is robust and not specific to the chosen threshold of t=0.25?

4) Why was EGM (Li et al., 2025a) not included as a comparison baseline in Table 1, given that it is a closely related method from the same group and that the FEF module is explicitly motivated by EGM's design? Including this comparison would clarify the marginal gain of the explain-by-design approach.

5) The dataset is heavily skewed toward MHC-I (95.7%). Do the results hold for MHC-II interactions? Given the stated relevance to autoimmune disease (where CD4+ T cells and MHCII are central), a breakdown of performance by MHC class would be valuable.


## Limitations

The authors do not include a formal limitations section, though the conclusion briefly mentions future directions. The following limitations are not adequately addressed in the paper:

- The theoretical justification for using embedding-space cosine similarity as a contact proxy is absent
- The evaluation relies on a single case study for structural validation
- Results are predominantly for MHC-I, leaving MHC-II generalization unclear
- The contribution of the FEF module versus the contact prototype layers is not disentangled
- Negative sample generation by random shuffling may introduce dataset-level biases that could artificially inflate performance metrics

---

## Minor Comments

- In Section 3.2, the text states "For peptide to CDR3 interactions, all backbones with contact prototype layers achieve BRHR values above 0.73" and then immediately repeats "For peptide to CDR3 interactions, BRHR exceeds 0.83 across all PLMs", this appears to be a copy-paste error referring to the same direction twice. One sentence likely refers to CDR3->Peptide interactions. Please clarify.
- The caption of Figure 1 states "In the absence of experimental TCR-pMHC structures, the contact prototype illuminates TCR-pMHC binding patterns." This is an important claim that would benefit from further elaboration in the main text, specifically, how the model behaves when no structural data is available at training time.
- Table 1 reports ROC-AUC values to three decimal places (e.g., 0.999), but does not report confidence intervals or standard deviations. Given the variability typical in TCR-pMHC benchmarks, reporting uncertainty estimates would strengthen the quantitative claims.
- The ROC-AUC of 0.999 for ProteinBERT at Top-100 appears extremely high. While this may be genuine, a brief discussion of why ProteinBERT outperforms ESM-1b and ESM-2 would be informative and help readers calibrate expectations.

## Overall Assessment

TCR-EML is a well-motivated, technically sound, and practically valuable contribution to the TCR-pMHC binding prediction literature. The explain-by-design framing is timely and appropriate; the contact prototype formulation is biologically grounded; and the results demonstrate consistent, substantial improvements over existing methods in both predictive accuracy and explainability. The paper is clearly written and complete.

The primary weaknesses are the lack of ablation separating the FEF and prototype contributions, the limited structural validation (single case study), and the absence of a comparison with the closely related EGM baseline. These are meaningful gaps but addressable, and do not undermine the core contribution. For a workshop venue, the work represents a mature and polished early contribution that the LMRL community would benefit from hearing.

---

### Official Review · Reviewer_irKH · 2026-02-25
**Plug-and-play explainability for protein language model binding interactions.**

**Rating:** 8
**Confidence:** 3

**Review:**

### Summary —

This work presents a method of adapting protein language models (PLMs) so that their output is interpretable. The philosophical motivation for this study can be attributed to Rudin (2019), that black-box models should not be used in high-stakes applications, which is especially pertinent in the immunological context of this work. The goal of this adaption is to score binding interaction.

Using an output head that is PLM agnostic, three embeddings are extracted from the backbone model, and using repeated cross attention that is biochemically inspired, predict TCR-pMHC binding.

The FEF block takes the PLM output and shares embedding information in a biochemically sound order. Then, the contact prototype block takes the enhanced embeddings and computes a similarity matrix $S$. The output is the average of two physical contact scores for CDR3a and CDR3b.

### Strengths —

- Explores a critically important piece of drug development via a thoughtful XAI approach.
- Plug-and-play functionality makes this easy to adapt to PLMs.
- No need to retrain the PLM backbone.

### Weaknesses —

I’m somewhat confused by the filtering/thresholding step for $S$ using $T$. Are these thresholds fixed? Wouldn’t they vary for the backbone?

### Questions for authors —

Some explanation or application of this method would strengthen the motivation of this paper. That is, beyond philosophical motivations, what would be an ideal use case of XAI in this domain of immunology? The conclusion mentions regularization as a use case. That is, would you add some additional regularization term to the loss to make minimization more biochemically informed? Additionally, how might this inform a clinician's workflow? I.e how would the score information be used in a drug development pipeline?

Notes:

Figures 1 & 2: The asterisk notation on FEF could be misread as indicating frozen weights. Figure 1 is missing the label entirely.

L61 - the acronym PLM should be written out.

---

### Meta-Review · Area_Chair_GUfy · 2026-02-28

**Recommendation:** Accept (Poster)
**Confidence:** 4

**Metareview:**

The reviewers mostly agree that this is a strong contribution, especially for the tiny track. Reviewer 3kJg raised some valid concern about the metics which I'd recommend the authors address before submitting a full version of the paper somewhere, but this clearly meets the threshold for LMLR.

---

### Decision · Program_Chairs · 2026-03-02

**Decision:**

Accept (Spotlight)

**Comment:**

Please see the meta-review.